# Cannabidiol inhibits Na$_v$ channels through two distinct binding sites

Jian Huang[1,4], Xiao Fan[1,4], Xueqin Jin[2], Sooyeon Jo[3], Hanxiong Bear Zhang[3], Akie Fujita[3], Bruce P. Bean[3] ✉ & Nieng Yan ◉[1,2] ✉

Cannabidiol (CBD), a major non-psychoactive phytocannabinoid in cannabis, is an effective treatment for some forms of epilepsy and pain. At high concentrations, CBD interacts with a huge variety of proteins, but which targets are most relevant for clinical actions is still unclear. Here we show that CBD interacts with Na$_v$1.7 channels at sub-micromolar concentrations in a state-dependent manner. Electrophysiological experiments show that CBD binds to the inactivated state of Na$_v$1.7 channels with a dissociation constant of about 50 nM. The cryo-EM structure of CBD bound to Na$_v$1.7 channels reveals two distinct binding sites. One is in the IV-I fenestration near the upper pore. The other binding site is directly next to the inactivated "wedged" position of the Ile/Phe/Met (IFM) motif on the short linker between repeats III and IV, which mediates fast inactivation. Consistent with producing a direct stabilization of the inactivated state, mutating residues in this binding site greatly reduced state-dependent binding of CBD. The identification of this binding site may enable design of compounds with improved properties compared to CBD itself.

Cannabidiol (CBD) is a major phytocannabinoid present in cannabis[1]. Unlike delta-9-tetrahydrocannabinol, the main psychoactive phytocannabinoid, CBD does not activate CB1 or CB2 cannabinoid receptors and does not show intoxicating effects. Nevertheless, CBD has clear effects on neuronal function. Large clinical trials have shown the efficacy of CBD for treating several childhood epilepsies[2–6], for which it is now FDA-approved. CBD has also been shown to relieve pain in animal models[7–9], as well as in several small clinical trials[10,11].

CBD has been shown to interact with a huge variety of proteins[12,13], especially membrane proteins[14], and it is still unclear which of these are the most important targets for CBD's action on epilepsy or pain. One of the actions of CBD is to inhibit voltage-gated sodium (Na$_v$) channels[15–19]. As for many drugs that inhibit Na$_v$ channels, the interaction of CBD with Na$_v$ channels is state-dependent, with higher affinity binding to the closed inactivated state of the channel than the

closed resting state[17]. Because Na$_v$ channels exist in a steeply voltage-dependent equilibrium between resting and inactivated states at physiological membrane potentials, tight binding to inactivated channels decreases the pool of resting state channels available for activation by a coupled-equilibrium mechanism, even if the drug binding is to a closed state of the channel[20,21].

Classic mutagenesis experiments revealed a key site for state-dependent interactions of local anesthetics like lidocaine and anti-seizure drugs like carbamazepine and phenytoin with Na$_v$ channels, formed by residues in the pore-lining S6 segments in the upper region of the pore[22–24]. Subsequent determinations of the structure of both bacterial and eukaryotic Na$_v$ channels revealed the presence of "fenestrations" in the side of the pore near this region[25–29], suggesting a pathway by which hydrophobic molecules can access the binding site from the lipid membrane – providing a concrete structural basis for the "hydrophobic pathway" hypothesized by Hille[20].

[1]Department of Molecular Biology, Princeton University, Princeton, NJ 08544, USA. [2]Beijing Frontier Research Center for Biological Structures, State Key Laboratory of Membrane Biology, Tsinghua-Peking Joint Center for Life Sciences, School of Life Sciences, Tsinghua University, Beijing 100084, China. [3]Department of Neurobiology, Harvard Medical School, 220 Longwood Avenue, Boston, MA 02115, USA. [4]These authors contributed equally: Jian Huang, Xiao Fan. ✉e-mail: bruce_bean@hms.harvard.edu; nyan@princeton.edu

Here we have used cryo-EM to determine the structure of CBD-bound Na$_v$1.7 channels. Unexpectedly, structures with and without CBD reveal two distinct binding sites for CBD. One is in fenestration in the upper pore. The other is a site close to where the IFM motif of the intracellular linker between domains III and IV[30] binds to produce the rapid time- and voltage-dependent inactivation characteristic of eukaryotic Na$_v$ channels by acting as a "door wedge" to squeeze the intracellular gate of the channel closed[28,29,31,32]. Mutating the residues identified in this binding site for CBD greatly reduced state-dependent CBD inhibition, suggesting a mechanism by which CBD binding stabilizes inactivation quite directly.

## Results

### CBD stabilizes the inactivated state of Na$_v$1.7

Although CBD is known to interact with a wide variety of proteins, substantial effects at sub-micromolar concentrations have been reported for only a few targets. Among these are Na$_v$1.8 channels[33], which are prominently expressed in primary nociceptors. We tested low concentrations of CBD on Na$_v$1.7 channels, the other major sodium channel driving excitability of primary nociceptors[34,35], using a stable cell line expressing human Na$_v$1.7 channels. Na$_v$1.7 current evoked from a holding potential of −70 mV, near the normal resting potential of nociceptors, was strongly inhibited by 300 nM CBD (Fig. 1a, b). In the collected results, the current was reduced to 0.29 ± 0.02 of control by 300 nM CBD applied for two minutes. The inhibition by 300 nM CBD was strongly sensitive to the state of the channels, as assayed by 5-second prepulses to alter the equilibrium between resting states and inactivated states of the channel (Fig. 1c). The midpoint of channel availability was shifted in the hyperpolarizing direction by an average

of 7.3 mV. This shift suggests tighter binding of CBD to inactivated states than resting states, and a simplified model of binding to resting and inactivated states[21] suggests a binding affinity of about 50 nM to inactivated states. One interpretation of the altered availability of Na$_v$ channels is that CBD stabilizes the inactivated state of the channel. Consistent with this, after a depolarizing prepulse to −40 mV to induce high-affinity CBD binding to inactivated states, recovery of availability is much slower than in control (Fig. 1d).

Like other Na$_v$ channels, Na$_v$1.7 channels can undergo two forms of inactivation, fast inactivation, and slow inactivation. With maintained depolarization, especially to strongly depolarized voltages, Na$_v$ channels can enter into a distinct slow inactivated state, from which recovery on repolarization is much slower. In principle, the high-affinity binding of CBD to inactivated states could involve either fast inactivated or slow inactivated states. Figure 1e shows an experiment designed to test whether CBD can bind to fast inactivated channels with high affinity, by examining the time course of entry of channels into slowly recovering states at −40 mV, where there is little slow inactivation. In the presence of CBD, there is substantial entry of channels into slowly recovering states within 20 ms at −40 mV, a time where there is almost no entry into slow inactivated states in control. This implies that CBD binds with high affinity to fast inactivated states. The shift of availability with long (5-s) depolarizations (Fig. 1c) suggests that CBD remains bound to channels in slow inactivated states, suggesting that whatever binding site is formed during fast inactivation remains when channels transition to slow inactivated states. The slowed recovery from inactivation following 300-ms depolarizations to −40 mV in the presence of CBD (Fig. 1d) could reflect either slow recovery of CBD-bound fast-inactivated channels or a larger

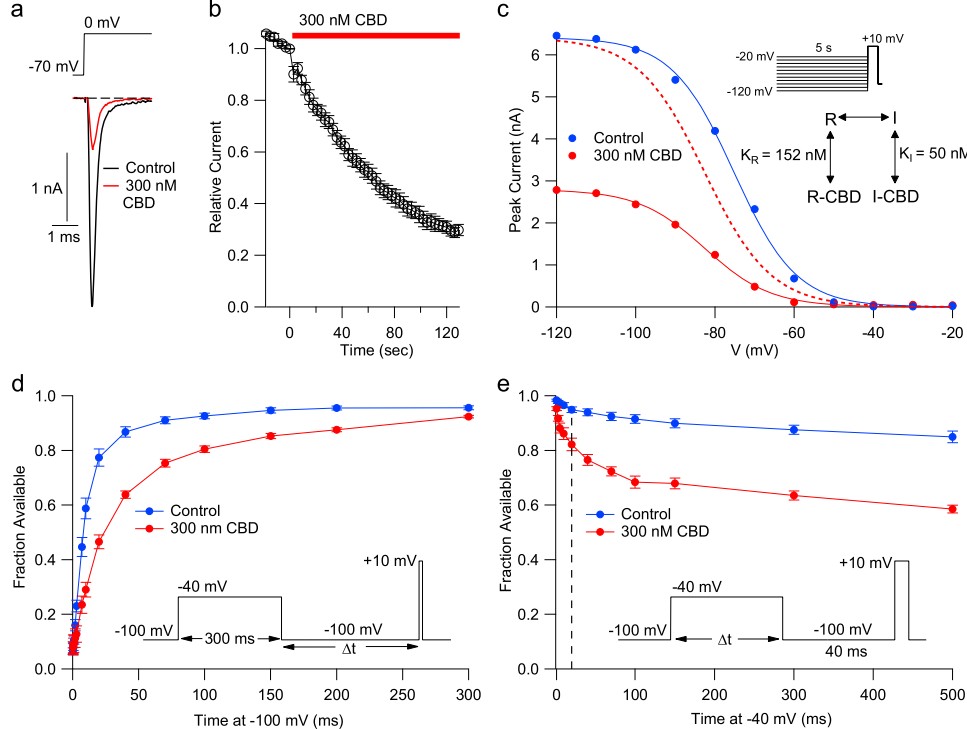

**Fig. 1 | Potent stabilization of Na$_v$1.7 inactivation by CBD. a** Currents before and after application of 300 nM CBD for two minutes at 37 °C. **b** Time-course of inhibition of hNa$_v$1.7 channels by 300 nM CBD at 37 °C. Mean ± SEM, $n$ = 7 cells. **c** Shift in the voltage-dependence of steady-state channel availability (determined by 5-s prepulses followed by a test pulse to +10 mV) by 300 nM CBD. Closed circles: Peak test pulse current versus prepulse voltage in a representative cell before and after 300 nm CBD. Solid lines: fits to Boltzmann function (control: midpoint −75.2 mV, slope factor 7.9 mV; 300 nM CBD: midpoint −82.6 mV, slope factor 8.4 mV). The dashed line shows the Boltzmann function fitted to CBD data normalized to the

control Boltzmann fit to show the shift in voltage dependence. Inset: calculated values for dissociation constants of CBD binding to resting and inactivated states[21], using average data from 6 cells. **d** Slowed recovery from inactivation with CBD, assayed at −100 mV following a 300-ms step to −40 mV to promote binding of CBD to fast inactivated channels (mean ± SEM, $n$ = 6 cells for recovery times ≥ 10 ms, 5–6 cells for times <10 ms). **e** Time-course of entry into slowly-recovering states during steps to −40 mV in the absence and presence of CBD (mean ± SEM, $n$ = 6 cells). A dashed line is drawn at 20 ms, showing rapid entry of channels into slowly-recovering states consistent with binding to fast inactivated states.

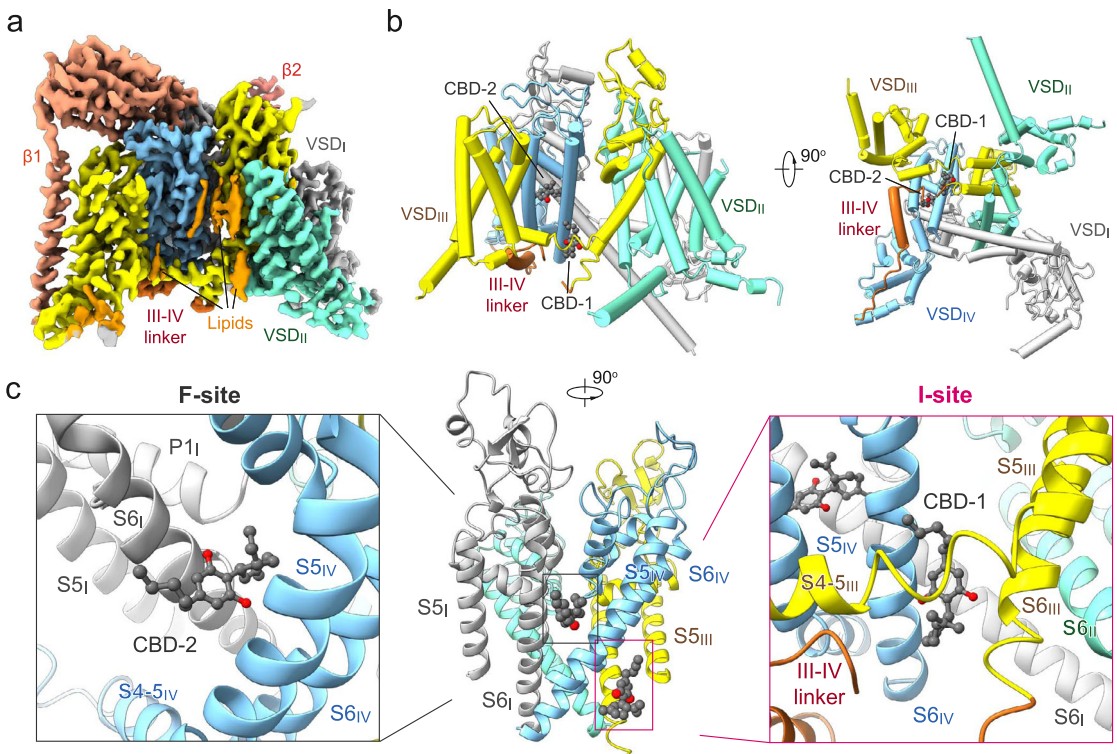

**Fig. 2 | Cryo-EM structure of the human Na$_v$1.7-CBD complex. a** Cryo-EM map of human Na$_v$1.7 bound to CBD. The Na$_v$1.7 complex comprises the transmembrane α1 subunit (domain colored), the auxiliary β1 subunit (light salmon), and β2 subunit (salmon). Sugar moieties and lipids are colored gray and orange, respectively. The same color scheme is applied throughout the manuscript. **b** Overall structure of the α1 subunit of Na$_v$1.7 bound to CBD. Two CBD molecules, designated as CBD-1 and CBD-2 and shown as light grey spheres, bind to different sites in the pore domain (PD). **c** Two CBD-binding sites, the inactivation receptor site (the I-site, *right*) for CBD-1 and the I-IV fenestration site (the F-site, *left*) for CBD-2, are identified and highlighted with magenta and gray boxes, respectively.

fraction of channels in the slow inactivated state or a combination of the two.

## CBD occupies two distinct sites on the pore domain (PD)

To examine the structural basis of CBD binding, we obtained a cryo-EM structure of CBD bound to human Na$_v$1.7 channels by including CBD throughout the purification of the channel protein starting with the first step after cells were harvested. Following our standard protocol for human Na$_v$1.7 purification and cryo-EM analysis, a 3D EM reconstruction of Na$_v$1.7-β1-β2 in the presence of CBD was obtained at 2.8 Å resolution (Fig. 2a, Supplementary Figs. 1–3 and Supplementary Table 1).

Map comparison with the ligand-free Na$_v$1.7 (EMDB: EMD-32368) immediately reveals one extra stretch of density that is right next to the IFM motif. In addition, a linear density that penetrates the IV-I fenestration in apo-Na$_v$1.7 is now replaced by a bulkier one in the presence of CBD. Both densities can be docked well with CBD (Fig. 2b and Supplementary Fig. 3a). This identifies two distinct CBD binding sites from the high-resolution EM map. For description simplicity, we name the site next to the IFM motif as the I-site for CBD-1 and the IV-I fenestration as the F-site for CBD-2 (Fig. 2c). In the following, we will present a detailed analysis of the two binding sites.

## CBD reshapes the IFM-binding site

The receptor site for the fast inactivation motif IFM in eukaryotic Na$_v$ channels is constituted mainly by hydrophobic residues from S4-S5$_{III}$, S5$_{III}$, S6$_{III}$, S5$_{IV}$, and S6$_{IV}$. Prominent conformational changes occur to the intracellular half of S6$_{III}$ upon CBD binding. Minor shifts are also observed in the adjacent segments (Fig. 3a). Although the position of the IFM motif remains unchanged, S6$_{III}$ is pushed away by CBD-1. More dramatically, residues 1461-1465, which fold in two helical turns on the

intracellular terminus of S6$_{III}$ in the ligand-free structure, are unwound to a loop in the presence of CBD-1, and the ensuing five residues become invisible (Fig. 3b, Supplementary Fig. 4a). Despite the marked shift of S6$_{III}$, the intracellular gate remains in a non-conducting state.

CBD-1, with the substituted cyclohexene rings on the intracellular side of the membrane and the *n*-pentyl tail hooking on the S4-S5$_{III}$ segment, is immersed in a largely hydrophobic environment encompassed by residues on S4-S5$_{III}$, S5$_{III}$, S6$_{III}$, S5$_{IV}$ and S6$_{IV}$, as well as Ile1472 in the IFM motif (Fig. 3c). Ser1320 and Asn1459 each form a hydrogen bond with the phenolic hydroxyl group of CBD-1 (Fig. 3d, Supplementary Fig. 5a).

To test whether the I-site is involved in CBD stabilization of inactivated states, we examined whether Na$_v$1.7 variants that contain either single point mutations (S1320A, N1459A) or a double mutation combining S1320A and N1459A alter the CBD-induced shift in availability reflecting high-affinity binding to inactivated states. All of the single and double mutations of the identified CBD-1 binding site substantially reduced both the sensitivity of Na$_v$1.7 to CBD and the shift in availability, and similar reductions were seen using both 5-s prepulses to produce steady-state inactivation or 50-ms prepulses to emphasize fast inactivation (Fig. 3e, f, Supplementary Figs. 7a, 8 and Supplementary Tables 2–3). These results suggest that the I-site makes a major contribution to the stabilization of the fast inactivated state by CBD.

## CBD binds to the IV-I fenestration site in a conserved pose

The F-site for CBD-2 bound in the IV-I fenestration is mainly composed of hydrophobic residues from S6$_I$, S5$_{IV}$, S6$_{IV}$, and P1$_{IV}$ (Fig. 4a, b and Supplementary Fig. 3).

The phenolic hydroxyl group of CBD is hydrogen-bonded by the backbone carbonyl group of Val383, and the phenolic ring is further

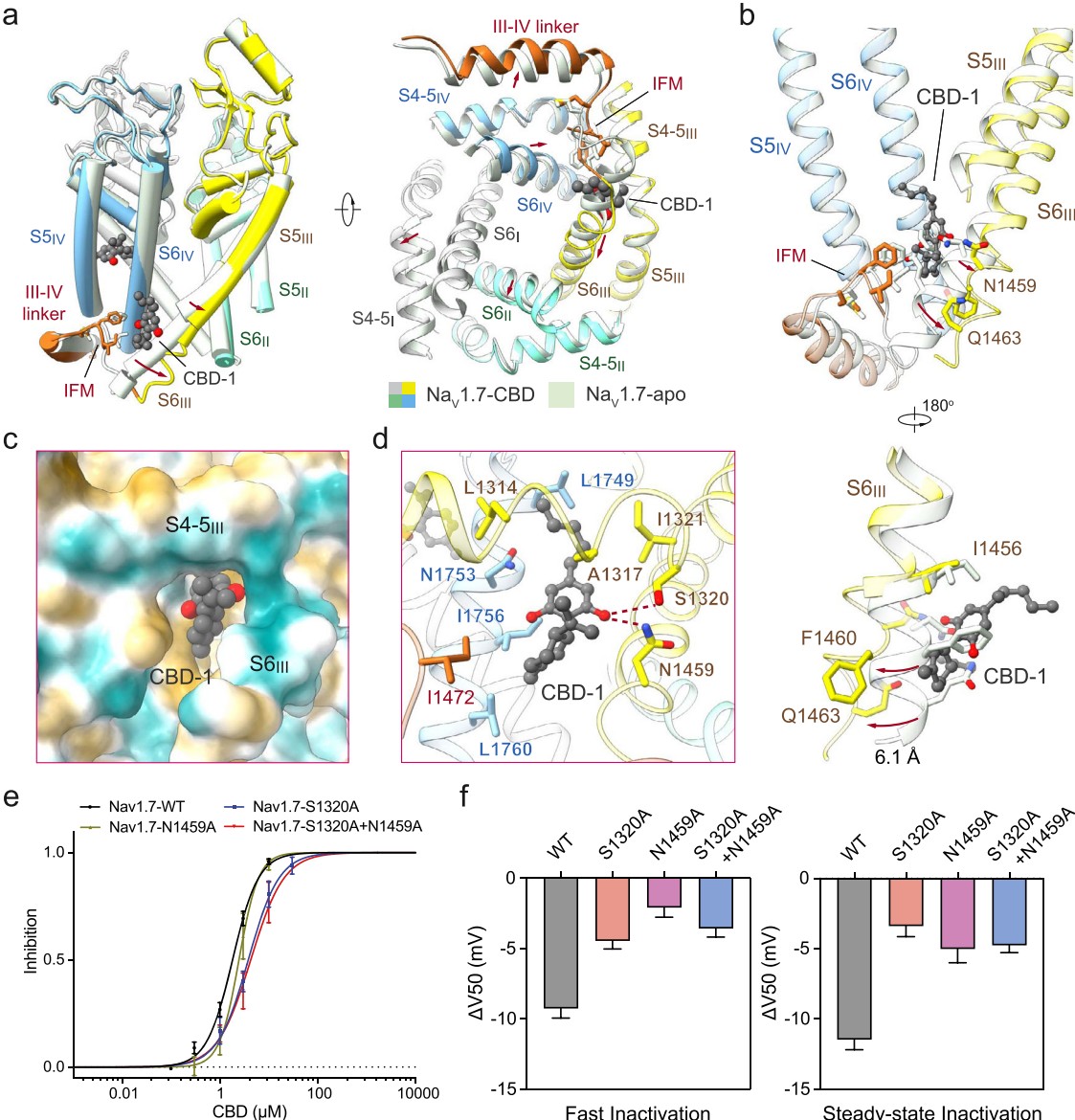

**Fig. 3 | Conformational changes of Na_v1.7 upon CBD binding to the I-site. a** CBD binding to the I-site induces pronounced conformational shift of the PD. A side view (*left*) and a bottom view (*right*) of the superimposed PD of CBD-bound (domain colored) and apo (pale green, PDB: 7W9K) Na_v1.7 are shown. CBD molecules are shown as grey spheres. The IFM motifs are shown as sticks, and the conformational changes are indicated with red arrows. **b** Rearrangement of the IFM binding site and surrounding elements upon CBD binding. Displacement of the corresponding residues upon CBD binding is indicated by red arrows. An enlarged view of the conformational shift of S6_III is shown at the bottom. **c** CBD binds to a hydrophobic pocket in the I-site. The surrounding environment is shown as the hydrophobic surface, calculated in ChimeraX[65]. **d** CBD coordination in the I-site. Surrounding residues are shown as sticks. Potential hydrogen bonds are shown as red dashed lines. **e** Mutations at the I-site modify CBD inhibition. Two single point mutations S1320A and N1459A and a double mutation S1320A/N1459A reduced the sensitivity of Na_v1.7 to CBD, with the IC_50 shifted from $1.82 \pm 0.10\ \mu M$ to $3.81 \pm 0.42\ \mu M$, $2.46 \pm 0.28\ \mu M$, and $4.28 \pm 0.67\ \mu M$, respectively. Na_v1.7-WT, $n = 1, 5, 12, 8, 7$. Na_v1.7-S1320A, $n = 3, 5, 5, 2$. Na_v1.7-N1459A, $n = 3, 9, 7, 5$. Na_v1.7-S1320A + N1459A, $n = 7, 7, 4$. **f** Mutations at I-site residues modify shifts in fast (50-ms prepulses) and steady-state inactivation (5-s prepulses) induced by $1\ \mu M$ CBD. The $\Delta V_{50}$ values for fast inactivation: $-9.26 \pm 0.69\ mV$ (WT), $-4.46 \pm 0.57\ mV$ (S1320A), $-2.09 \pm 0.69\ mV$ (N1459A), and $-3.58 \pm 0.61\ mV$ (S1320A + N1459A); for steady-state inactivation: $-11.46 \pm 0.72\ mV$ (WT), $-3.39 \pm 0.75\ mV$ (S1320A), $-5.02 \pm 0.98\ mV$ (N1459A) and $-4.76 \pm 0.52\ mV$ (S1320A + N1459A). Data represent mean $\pm$ SEM. *n* biological independent cells.

stabilized by Phe387 through a π-π stacking interaction. The lipophilic tail interacts only with the side chain of Val383 through van der Waals interaction (Fig. 4b, Supplementary Fig. 5b). Single point mutations at both V383A and F387A reduced the inhibitory activity of CBD on Na_v1.7, supporting the structural observation of this binding site (Fig. 4c, Supplementary Fig. 7b and Supplementary Table 2).

The structure of the F-site in the IV-I fenestration remains nearly unchanged with or without CBD (Fig. 4b, Supplementary Fig. 4b). Interestingly, the IV-I fenestration is absent in the structure of Na_v1.7 bound to saxitoxin and the gating modifier Huwentoxin-IV[36]

(Supplementary Fig. 4b), suggesting that the fenestration may be dynamically regulated in different gating states.

Previous work determined a cryo-EM structure of rat TRPV2 treated with CBD[37]. In the TRPV2 structure, each of the four fenestrations is occupied by a CBD molecule. Unlike in the homotetrameric TRPV2 channel, CBD recognizes only one fenestration enclosed by repeats I and IV in the pseudosymmetric Na_v1.7 (Fig. 4d). Notably, CBD shares a similar binding pose in Na_v1.7 and rTRPV2 at the F-site, with the cyclohexene ring pointing towards the central cavity and the *n*-pentyl tail exposed to the membrane (Fig. 4d).

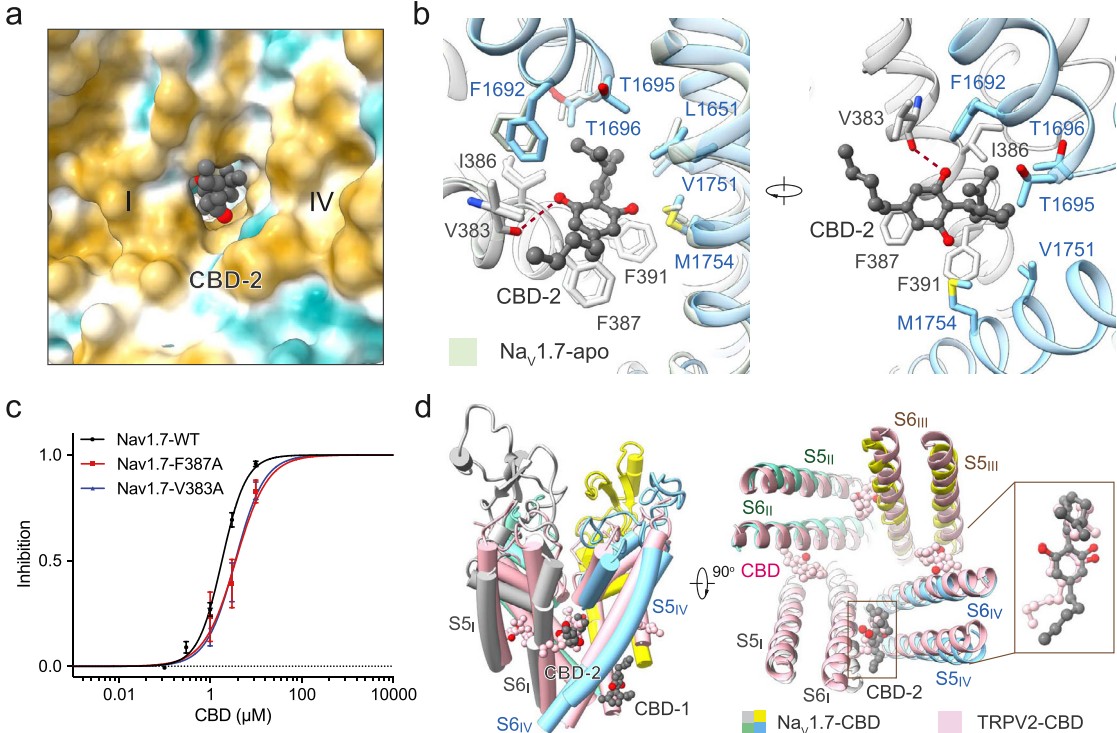

**Fig. 4 | Coordination of CBD at the F-site. a** CBD binds to the fenestration enclosed by repeats IV and I of Na$_v$1.7. The surrounding environment, which is highly hydrophobic, is shown as the hydrophobic surface, calculated in ChimeraX. **b** The local structures around the F-site (*left*) remain nearly identical in apo (PDB: 7W9K) and CBD-bound Na$_v$1.7. Surrounding residues are shown as sticks. The only potential hydrogen bond is shown as red dashed lines. **c** F-site mutations weaken the inhibition of Na$_v$1.7 by CBD. Both single point mutations V383A and F387A reduced the sensitivity of Na$_v$1.7 to CBD, with the IC$_{50}$ right-shifted from

1.82 ± 0.10 μM to 3.56 ± 0.58 μM and 3.65 ± 0.78 μM, respectively. Nav1.7-WT, $n = 1$, 5, 12, 8, 7. Nav1.7-F387A, $n = 4$, 6, 4. Nav1.7–V383A, $n = 5$, 4,5. Data are presented as mean ± SEM. *n* biological independent cells. **d** CBD shares similar binding poses in Na$_v$1.7 and TRPV2 at the F-site. A similar fenestration binding site for CBD in Na$_v$1.7 and in TRPV2 is seen in the superimposed structures. A side view (*left*) and a top view (*right*) of the comparison between CBD-bound Na$_v$1.7 (domain colored) and TRPV2 (pink, PDB: 6U88) are shown.

A structure of the bacterial channels with bound CBD has also been determined[38]. CBD in Na$_v$MS is also present in fenestrations. However, the binding pose in the fenestrations does not overlap with that in Na$_v$1.7, with the molecule extending much deeper into the central cavity in the bacterial channel (Supplementary Fig. 9).

## Discussion

Unexpectedly, CBD binds to two different sites on Na$_v$1.7 channels, and both sites are different from the binding sites of other small-molecule state-dependent Na$_v$ inhibitors previously determined from structural data with mammalian Na$_v$1 channels, including flecainide[39], quinidine[32], propafenone[40], bulleyaconitine A[41], XEN907, TC-N1752[42], and A-803467[43]. Neither of the two CBD molecules bound to Na$_v$1.7 channels is in a position to physically occlude the pore. This suggests that CBD acts allosterically to inhibit Na$_v$1.7 channels, and the high-affinity binding to the inactivated state of the channel shown by the electrophysiological experiments suggests that the key allosteric action is a stabilization of the inactivated state. The position of the I-site for CBD-1 binding at the receptor site for the IFM motif, and the dramatic changes in the S6$_{III}$ structure with CBD, suggest that CBD binding at this site may stabilize inactivation very directly, by increasing the binding of the IFM motif, resulting in both a shift of the voltage-dependent equilibrium between resting and inactivated channels and a slowing of recovery from inactivation when the membrane is repolarized.

CBD is an exceptionally lipophilic molecule. With a logP of 6.33, it is expected to be ~ 10$^6$ times more concentrated in the lipid membrane than in the aqueous extracellular or intracellular solutions. The discovery of fenestrations in Na$_v$ channels of both prokaryotes and

eukaryotes led to the hypothesis that the movement of hydrophobic molecules through the fenestrations could form the basis for the "hydrophobic pathway" hypothesized by Hille[20] to account for the differing kinetics of development and recovery from channels inhibition by charge or uncharged local anesthetics. The idea that the fenestrations can form pathways for effective movement of hydrophobic molecules between the lipid membrane and the channel pore has been well-supported by both modeling and experiments[44–46]. The presence of a CBD molecule in the IV-I fenestration is consistent with the idea that CBD access to the channel protein from its high concentration in the membrane can occur in part through this fenestration. Interestingly, in the structure of CBD-bound homotetrameric bacterial Na$_v$ channel Na$_v$Ms, CBD molecules were modeled in each of the four fenestrations to extend into the central cavity of the pore. However, the CBD molecule in the IV-I fenestration of Na$_v$1.7 does not extend into the pore cavity and seems unlikely to directly block the pore. More work will be needed to determine whether the binding of the CBD molecule at the F-site helps stabilize the inactivated state of the channel by an indirect allosteric mechanism.

The binding of the CBD-1 to the I-site may not require movement through a fenestration, because the I-site is located outside of the pore domain, and thus, may be directly accessible from the lipid bilayer. The identification of the two binding sites for CBD in Nav1.7 should help refine models of movements of CBD between the extracellular solution, the lipid bilayer, and the channel (e.g. ref. 47). It is notable that the effect of CBD, although very potent, is also very slow, taking several minutes to approach steady-state (Fig. 1a).

Previous studies of lidocaine, carbamazepine, and other small molecule Na$_v$ channel inhibitors identified a key phenylalanine residue

in the upper region of the pore whose mutation dramatically reduces the inhibition by those compounds[22,24]. Mutation of this residue in both Na$_v$1.1[17] and Na$_v$1.4[47] had much smaller effects on the potency of CBD inhibition compared to large effects on the potency of carbamazepine and tetracaine. Our results support the conclusion from these experiments that CBD likely acts primarily through different binding sites. Similar to the results with Na$_v$1.1[17] and Na$_v$1.4[47] we found that mutating the phenylalanine crucial for local anesthetic action in Na$_v$1.7 produced some reduction in the action of CBD (Supplementary Fig. 11) but the effect was small compared to the effect of mutations at either the I-site or F-site. Because there is no evident binding of CBD directly to this residue, the effects of mutations at this residue could occur through an indirect allosteric mechanism.

Many small molecule Na$_v$ channel inhibitors, including other antiepileptic drugs and local anesthetics, share the property with CBD of binding more tightly to the inactivated state of the channel than the resting state, even though we now see that the binding sites are quite different[48]. The state-dependent binding to inactivated channels results in inhibition that is more pronounced when the neuronal membrane is more depolarized, typically associated with higher firing rates, and is likely the reason that sodium channel-targeted antiepileptic agents can reduce hyperactivity without debilitating effects on normal activity. Such selectivity is likely also desirable for pain management since total loss of pain has devastating consequences. Although CBD shares the property of state-dependence with many other clinically useful drugs, the calculated binding affinity of ~ 50 nM for CBD binding to inactivated channels is much higher than for any other anti-epileptic drug with similar state-dependent binding characteristics (e.g. ~ 7 μM for phenytoin[49]; ~9 μM for lamotrigine[50], 25 μM for carbamazepine[51], and ~14 μM for lacosamide[52]) or local anesthetics like lidocaine[21] (~20 μM).

Despite its clinical use for epilepsies that cannot be controlled by other agents, cannabidiol has major limitations as a drug, including inducing somnolence, vomiting, diarrhea, and hepatic abnormalities[53]. Many of these adverse effects likely reflect the promiscuity of CBD in affecting so many proteins not likely to be involved in its beneficial actions. The knowledge of the binding sites for CBD on Na$_v$ channels should facilitate the structure-based discovery of drugs deliberately targeted to sodium channels with better specificity.

## Methods
### Transient expression of human Na$_v$1.7 in HEK293F cells
Codon-optimized cDNA for full-length human Na$_v$1.7 (Uniprot Q15858), a gift from Tsinghua University, was cloned into the pCAG vector with Twin-Strep-tag and Flag-tag in tandem at the amino terminus while codon-optimized cDNAs for β1 subunit (Uniprot Q07699) and β2 subunit (Uniprot O60939) were cloned separately into the pCAG vector without affinity tag. All the plasmids for transient expression were verified by DNA sequencing. HEK293F suspension cells (Thermo Fisher Scientific, R79007) were cultured in SMM 293T-II medium (Sino Biological Inc.) at 37 °C, supplied with 5% CO$_2$ under 60% humidity, and transfected with plasmids when the cell density reached 1.5–2.0 × 10$^6$ cells per ml. For one-liter cell culture, a mixture of expression plasmids including 1.5 mg plasmids for Na$_v$1.7, 0.5 mg plasmids for β1, and 0.5 mg plasmids for β2 was pre-incubated with 4 mg 40-kDa linear polyethylenimines (Polysciences) in 50 ml fresh medium for 15–30 minutes, and then added to the cell culture for transient expression of human Na$_v$1.7 complex.

### Protein purification of human Na$_v$1.7-CBD complexes
A total of 33 L transfected HEK293F cells were harvested approximately 48 h after transfection by centrifugation at 3,600 g for 10 min and resuspended in the lysis buffer containing 25 mM Tris-HCl (pH 7.5) and 150 mM NaCl. The suspension was supplemented with 10 μM CBD (Sigma-Aldrich) and protease inhibitor cocktail (Selleckchem), and

incubated at 4 °C for 30 min. Then n-dodecyl-β-D-maltopyranoside (DDM, Anatrace) was added to a final concentration of 1% (w/v), and cholesteryl hemisuccinate Tris salt (CHS, Anatrace) to 0.1% (w/v). After incubation at 4 °C for another 2 h, the mixture was centrifuged at 16,000 g for 45 min, and the supernatant was applied to anti-Flag M2 affinity gel (Sigma) for affinity purification. The resin was rinsed four times with the wash buffer (buffer W) that contains 25 mM Tris-HCl (pH 7.5), 150 mM NaCl, 0.06% GDN, 10 μM CBD, and a protease inhibitor cocktail. The target proteins were eluted with buffer W supplemented with 0.2 mg mL$^{-1}$ Flag peptide (synthesized by GenScript). The eluent was then applied to Strep-Tactin Sepharose (IBA) and flew through by gravity. The resin was rinsed four times with buffer W and the target proteins were eluted with buffer W supplemented with 2.5 mM desthiobiotin (IBA). The eluent was then concentrated using a 100-kDa cut-off Amicon filter unit (Millipore) and further purified through size-exclusion chromatography (Superose 6 10/300 GL, GE Healthcare) that was pre-equilibrated in the buffer containing 25 mM Tris-HCl (pH 7.5), 150 mM NaCl, 0.02% GDN and 10 μM CBD. The peak fractions were pooled and concentrated to a final concentration of ~ 9 mg mL$^{-1}$ and incubated with 100 μM CBD at 4 °C for another 30 min before making cryo-girds.

### Cryo-EM sample preparation and data acquisition
UltrAuFoil (R1.2/1.3 300 mesh, Quantifoil) grids were glow-discharged with easiGlow (PELCO) using 15 mA for 15 s at 0.37 mBar. The Vitrobot Mark IV chamber was pre-cooled to 10 °C with 100% humidity. Three microliters of concentrated Na$_v$1.7-CBD was applied to the freshly treated grid surface, which was then blotted with filter paper for 4 s and plunged into liquid ethane cooled by liquid nitrogen. Grids were loaded to a 300 kV Titan Krios G3i with spherical aberration (Cs) image corrector (Thermo Fisher). SerialEM[54] was used for automated data collection of both no-tilt and tilted micrographs. Micrographs were recorded by a Gif Quantum K2 Summit camera (Gatan) with 20 eV slit in super-resolution mode at a nominal magnification of X105,000, resulting in a calibrated pixel size of 0.557 Å. Each movie stack was exposed for 5.6 s (0.175 s per frame, 32 frames) with a total electron dose of ~ 50 e$^-$/Å$^2$. The movie stacks were aligned, summed and dose-weighted using Warp[55] and binned to a pixel size of 1.114 Å per pixel.

### Data processing
In total 5,862 cryo-EM micrographs were collected from four subsets of micrographs, including two no-tilt subsets (1$^{st}$: 1302; 2$^{nd}$: 2286), a 30° tilted subset (3$^{rd}$: 1271) and a 40° tilted subset (4$^{th}$: 1003). Warp[55] preprocessed subsets were imported to cryoSPARC[56] for patched CTF estimation. 41,155 particles from 130 micrographs were auto-picked by blob picking to generate 2D templates through 2D classification. 21,839 particles in 15 classes were selected to perform Ab-initio reconstruction and the class averages were used in later template picking for all subsets. Each subset was picked individually with selected templates, yielding 783,670/1,336,960/1,292,889/1,712,983 particles for the 4 subsets. Patch CTF extraction jobs were performed to update local defocus information in both 30° and 40° tilted datasets based on particle coordinates. Obvious junks were excluded from extracted bin4 particles in each subset by 2D classification. Those roughly cleaned subsets were sent to heterogenous refinement with 2 references generated from Ab-initio reconstruction. Bin2 particles were extracted with selected good classes and merged for the 4 subsets. After four rounds of heterogenous refinement and remove duplicates, 590,967 particles in the best class were left and were extracted into bin1. Those particles were subject to another two rounds of heterogenous refinement using information from higher frequency. After non-uniform (NU) refinement, a good 3D class containing 488,974 particles yielded a reconstruction at an overall resolution of 3.0 Å. These results were exported to Relion[57] to perform the Bayesian polishing with the motion parameters estimated by Warp.

Polished particles were re-imported to cryoSPARC to perform the NU refinement, which leads to a 2.8 Å final reconstruction.

## Model building and refinement

The coordinates of WT apo-Na$_v$1.7 (PDB: 7W9K), were used as the initial model for model building for human Na$_v$1.7-CBD. The published model was docked and saved related to the Na$_v$1.7-CBD EM density in Chimera[58]. The refitted initial model was adjusted in COOT first[59]. It was then refined against the corresponding map using the Real-space Refinement option in PHENIX[60] with secondary structure and geometry restraints. Validation for the model refinement can be found in Supplementary Table 1.

## Whole-cell electrophysiology

Experiments characterizing the state-dependent actions of CBD on wild-type hNa$_v$1.7 channels were made using HEK293 cells stably expressing human Na$_v$1.7 channels[61]. Cells were grown in Minimum Essential Medium (ATCC) containing 10% fetal bovine serum (Sigma) and 1% penicillin/streptomycin (Sigma) under 5% CO$_2$ at 37 °C. For electrophysiological recording, cells were grown on coverslips for 12 to 24 h after plating. Whole-cell recordings were obtained using patch pipettes with resistances of 2–3.5 MΩ when filled with the internal solution, consisting of 122 mM CsCl, 9 mM NaCl, 1.8 mM MgCl$_2$, 9 mM EGTA, 14 mM creatine phosphate (tris salt), 4 mM MgATP, and 0.3 mM GTP (tris salt), 9 mM HEPES, pH adjusted to 7.2 with CsOH. The shank of the electrode was wrapped with Parafilm to reduce capacitance and allow optimal series resistance compensation without oscillation. Recordings were made using a Multiclamp 700B amplifier (Molecular Devices) with currents and voltages controlled and sampled using a Digidata 1322 A interface using pClamp 9 software (Molecular Devices). Current and voltage records were filtered at 5–10 kHz and digitized at 100 kHz. Analysis was performed with Igor Pro (Wavemetrics, Lake Oswego, OR) using DataAccess (Bruxton Software) to import pClamp data. Sodium currents were corrected for linear capacitive and leak currents determined using 5 mV hyperpolarizations delivered from the resting potential (−70 or −100 mV) and then appropriately scaled and subtracted.

After establishing whole-cell recording in Tyrode's solution (155 mM NaCl, 3.5 mM KCl, 1.5 mM CaCl$_2$, 1 mM MgCl$_2$, 10 mM HEPES, 10 mM glucose, pH adjusted to 7.4 with NaOH) cells were lifted off the bottom of the recording chamber and placed in front of an array of quartz fiber flow pipes (250 μm internal diameter, 350 μm external diameter, Polymicro Technologies, Catalog # TSG250350) attached with styrene butadiene glue (Amazing Goop, Eclectic Products) to a rectangular aluminum rod (cross-section 1.5 cm × 0.5 cm) whose temperature was controlled by resistive heating elements and a feedback-controlled temperature controller (Warner Instruments, TC-344B). Solutions were changed (in ~ 1 second) by moving the cell from one pipe to another. Recordings were made at either 37 °C or 21 °C.

Sodium currents were recorded in an external solution consisting of 116 mM NaCl, 39 mM TEACl, 5 mM BaCl$_2$, 10 μM CdCl$_2$, 10 mM HEPES, 10 mM glucose, pH adjusted to 7.4 with NaOH. CBD (Cayman Chemical, Catalog # 90080, CAS 13956-29-1) was prepared as a 10 mM stock solution in DMSO which was diluted in the external solution to the final concentration. DMSO was added to the control solution at the same concentration as in the CBD solution. When using sub-micromolar concentrations of CBD, we used all-glass containers and perfusion tubing to avoid loss of compound by partitioning into plastic[62], which is problematic even for Δ$_9$-THC[63,64], which is less lipophilic (logP 5.65) than CBD (logP 6.3).

The effects of CBD on steady-state inactivation were determined using 5-s long prepulses to voltages between −130 mV and −20 mV followed by a 10-ms test pulse to +10 mV. The maximum voltage error remaining after partial compensation for series resistance (usually 70%) was calculated as the product of peak current and the series resistance remaining after compensation, and data were used only if the maximum voltage error was less than 12 mV; with this criterion, errors have minimal effect on voltage-dependence of inactivation when determined with a test pulse to +10 mV, just past the peak of the current-voltage relation, where changes in peak current with voltage are minimal. 300 nM CBD was applied for 10 minutes in order to reach a steady state. In control experiments, we found that there was a small time-dependent shift in the midpoint of inactivation over this time, and the average shift in control (with DMSO at the concentration present in the 300 nM CBD solutions) over 5 minutes (−3.6 mV, $n = 9$) was subtracted from the average shift with 300 nM CBD (−10.9 mV, $n = 6$) in order to calculate the CBD-induced shift that was used for fitting the state-dependent model in Fig. 1c. Data are given as mean ± SEM.

Experiments examining the effects of mutations used transient transfections in HEK293T cells. HEK293T cells were cultured in Dulbecco's Modified Eagle Medium (DMEM, BI) supplemented with 4.5 mg/mL glucose, 10% (v/v) fetal bovine serum (FBS, BI). For patch-clamp recordings, the cells were plated onto glass coverslips and transiently co-transfected with the Na$_v$1.7 variants plasmids and eGFP in the presence of lipofectamine 2000 (Invitrogen). Cells with green fluorescence were selected for patch-clamp recording 18–36 h after transfection. Experiments were performed at room temperature. No authentication was performed for the commercially available cell line. *Mycoplasma* contamination was not tested.

Whole-cell Na$^+$ currents were recorded in HEK293T cells using an EPC10-USB amplifier with Patchmaster software v2*90.2 (HEKA Elektronic), filtered at 3 kHz (low-pass Bessel filter) and sampled at 50 kHz. The borosilicate pipettes (Sutter Instrument) used in all experiments had a resistance of 2–4 MΩ; series resistance was compensated by 70–80% so that voltage errors were minimal with the typical current magnitudes of 1-2 nA. The electrodes were filled with the internal solution composed of 105 mM CsF, 40 mM CsCl, 10 mM NaCl, 1 mM EGTA, 10 mM HEPES, pH adjusted to 7.4 with CsOH. The bath solutions contained: 140 mM NaCl, 4 mM KCl, 10 mM HEPES, 10 mM D-Glucose, 1 mM MgCl$_2$, 1.5 mM CaCl$_2$, pH adjusted to 7.4 with NaOH. Data were analyzed using Origin (OriginLab) and GraphPad Prism (GraphPad Software).

To determine voltage dependence of inactivation, cells were clamped at a holding potential of −100 mV and pre-pulses from −130 mV to 0 mV for either 5 s (for steady-state inactivation) or 50 ms (for fast inactivation) were applied with an increment of 5 mV, followed by a 50-ms test pulse to 0 mV. Fast and steady-state inactivation curves were fitted with a Boltzmann function to obtain V$_{1/2}$ and slope values. Leak currents and capacitive transients were subtracted using the P/4 procedure. CBD was dissolved in dimethyl sulfoxide (DMSO, final concentration less than 0.1%, Sigma) to make a stock solution of 30 mM. Solutions with indicated CBD concentrations were freshly prepared and perfused to the recording cell for up to 10 mins to get to the maximal block using a multichannel perfusion system (VM8, ALA). Prior to CBD perfusion, cells were recorded for 5 min to establish a stable peak current. Data are presented as mean ± standard error of the mean (SEM) and n is the number of experimental cells from which recordings were obtained. Statistical significance was assessed using an unpaired t-test with Welch's correction and extra sum-of-squares F test.

Because the cryoEM structures were determined with channels that included β1 and β2 subunits, in addition to examining the effects of mutating the CBD binding sites determined from the structural data, we also tested whether the presence of β1 and β2 influenced the effect of CBD determined the electrophysiology experiments. As shown in Supplementary Fig. 10, the shift of availability with co-expression of β1 and β2 is very similar to that without beta subunits, suggesting that the interaction of CBD with the channel is little affected by the beta subunits.

**Reporting summary**

Further information on research design is available in the Nature Portfolio Reporting Summary linked to this article.

## Data availability

The data that support this study are available from the corresponding authors upon request. The cryo-EM map has been deposited in the Electron Microscopy Data Bank (EMDB) under accession code EMD-29665 (Na$_v$1.7-CBD). The coordinates have been deposited in the RCSB Protein Data Bank (PDB) under accession code 8G1A (Na$_v$1.7-CBD). Previously solved structures mentioned in this study are under the accession codes in PDB: 7W9K (Na$_v$1.7-apo), 6J8G (Na$_v$1.7-HWTX IV and STX), 6YZ0 (Na$_v$Ms F208L-CBD), and 6U88 (rTRPV2-CBD), respectively. The sequences of human Na$_v$1.7, β1 and β2 are available in the following links: Na$_v$1.7 (UniProtID:Q15858); β1 (UniProtID:Q07699); β2 (UniProtID:O60939). Source data are available as a Source Data File. Source data are provided with this paper.

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

## Acknowledgements

We thank the cryo-EM facility at Princeton Imaging and Analysis Center. Thanks to Dr. Clifford Woolf and Dr. Jed Hubbs for helpful discussion. Supported by the National Institutes of Health National Institute of Neurological Diseases and Stroke [NS036855, NS110860, NS127216] and the Charles R. Broderick III Phytocannabinoid Research Initiative. N.Y. was supported by the Shirley M. Tilghman endowed professorship from Princeton University in 2017-2022. X. F. has been supported by the HFSP long-term fellowship (LT000754/2020) from the International Human Frontier Science Program Organization (HFSPO).

## Author contributions

J.H., X.F., X.J, S.J., H.B.Z., A.F., B.P.B., and N.Y. designed experiments; J.H., X.F., X.J, S.J., H.B.Z., and A.F. carried out experiments; J.H., X.F., X.J, S.J., H.B.Z., A.F., B.P.B., and N.Y. analyzed data; J.H., X.F., X.J, B.P.B., and N.Y. wrote the manuscript with input from all authors. All authors approved the final manuscript.

## Competing interests

The authors declare no competing interests.
