## [Peer Review File · Nature Communications]

Cannabidiol inhibits Nav channels through two distinct binding sitesReviewers' Comments:

Reviewer #1:

Remarks to the Author:

The study is a very nice convergence of CryoEM experiments with functional, pharmacological ion channel research. It is fascinating and encouraging to see that methods like cryo-EM are being used as validation tools for experimental results. A physiologically highly relevant question was addressed using cutting edge technical approaches: how does cannabidiol relieve pain? It was shown before that it interacts with Nav1.8, a sodium channel known for its importance in human pain perception. Arguably even more important is Nav1.7, as humans lacking this channel cannot feel pain. The authors show in convincing patch clamp recordings that CBD enhances fast inactivation of Nav1.7. The binding sites are unclear, although the electrophysiological profile hints towards the local anesthetic binding site. To clarify this, the authors present the structure of Nav1.7 bound with CBD and find it in a rather surprising position: close to the IFM motive. Mutagenesis was used to explore this binding site and it was compared to a second binding position which is found closer to the permeation pore on S6 of Domain IV.

The topic is of high interest for both clinicians and basic scientists, as CBD gains more and more therapeutic relevance and the authors describe a new pharmacological binding site for voltage gated sodium channels. The manuscript is well written, the presentation of the data is appropriate. The relevance of each binding site for the channel gating should be investigated in more detail, and it would also allow to exploit the structural data to a greater extent.

Major:

1. The CryoEM structure shows binding of Beta1 and beta2. The ephys experiments on the other hand were performed in the absence of any beta subunit. Please show that the beta subunits do not interfere with/modify binding of CBD with Nav1.7.
2. Figure1: The ephys experiments look technically flawless, experiments were performed at 37°C, which is very nice and challenging. Part E: Despite pharmacologically altered recovery kinetics from fast inactivation, the observed slow recovery states of treated channels could also arise from slow inactivated states if the voltage dependence of slow inactivation was altered by CBD. Please discuss and explain, and/or add experimental data.
3. The authors show very interesting data on two binding sites of CBD. Especially the I-site is a so far unknown binding site for therapeutics. Therefore, a more detailed functional assessment of the structural-functional relation of each binding site would be desirable. It would be very enriching to further investigate the functional importance for each presented binding sites for the pharmacological effect on Nav1.7. E.g., the authors should use mutagenesis (e.g. to change the fenestration site size) to show the effect of a disruption of this binding site for the functional impact of CBD on the channel, or similar:
 - a. Concerning the lipophilic properties of CBD the F Site is likely the more occupied or do the authors assume that the I-side can be accessed via the fenestration pathway? If yes they could also use MD Simulations to investigate and further characterize this pathway.
 - b. The IV-I fenestration – close to the F-site - was suggested to be one of the smallest fenestrations [Tao and Corry, 2022]. It would be quite interesting and useful if the authors showed some analysis of the side fenestrations with and without the presence of CBD and also compare it to the other known Nav1.7 structures (7W9K, 6J8G).
 - c. The authors describe certain key residues such as F387 and V383, but do not show any mutagenesis experiments of these residues to determine whether CBD binding is affected (partially, completely or not at all). These may help to show whether the F-site close to the I-IV fenestration may have a higher affinity than or may even be dependent on binding of CBD on the I-Site.
 - d. An earlier study on Nav1.4 showed that the local anesthetic residue F1586 is important for CBD binding to sodium channels and that CBD directly occludes the pore by binding at the local anesthetic site near the DIII-IV fenestration [Ghovanloo et al., 2022]. This contrasts with the conclusion of this paper that CBD probably does not occlude the pore and binds in the IV-I fenestration and IFM binding site. To exclude this, experiments using a mutation of the local anesthetic binding site would help and should be easily performed.

e. The I-site is a very unorthodox site, given how the hydrophobicity of this pocket is considered crucial for proper binding of the motif. Did the authors consider checking for the hydrophobicity of the pocket with and without the drug to see if CBD cause any changes? This analysis should be added to the paper.

f. The authors mention that the F1463 of the IFM motif is in the pocket where CBD binds. Did the authors notice any pi-stack interactions between CBD and this residue? If so, it will help to perform experiments to test if CBD directly interacts with the IFM motif by mutating this residue e.g. to an alanine.

Minor:

1. Supplement 4: c and D please change scaling so that quick transient sodium traces become better visible.

2. Depending on the journal's rules, it may be preferable to use SD instead of SEM throughout the manuscript.

3. Methods: Ephys: No statement on series resistance or voltage error is given (and the pipettes are quite small in diameter). Please mention.

4. Figure 1: In Figure 1(c) the relative currents of CBD start from 0.4 but should start from (or close to) 1.

5. Figure 3(e) and Figure 4(a) inset: Residue interactions are shown only for hydrogen bonds and van der Waal's interactions, but the text in addition to these mentions pi-pi stack interactions which are missing in the figures. Please also show in the figures.

Reviewer #2:

Remarks to the Author:

The manuscript by Jian and colleagues investigated inhibitory mechanism of human voltage-gated sodium channel Nav1.7 by cannabidiol (CBD) by using structural and electrophysiological approaches. Through electrophysiological studies, the authors confirmed CBD can inhibit activity of Nav1.7 by specifically interacting with Nav1.7 in closed inactivated state rather than closed resting state. The high-resolution single particle cryo-EM complex structure of Nav1.7 with CBD bound at 2.8 Å revealed two distinct binding sites for CBD, one localized in the fenestration in the upper pore, the other close to IFM motif of the intracellular linker between domain III and IV. Structural analysis provided detailed interaction mode of CBD in Nav1.7, and subsequence mutagenesis and functional assay further prove roles of key residues in coordinating CBD. This high-quality work is another important contribution in mechanistic mechanism of the sodium channels field. By unraveling interacting details between Nav1.7 and CBD, this work provide value information for future drug improvement on the basis of CBD, and shed lights on treatment of diseases like epilepsies with little side effects.

Unexpectedly, there are two binding sites observed in the complex structure. These two binding sites differs from binding modes and effects on conformational rearrangement to regulate protein function. Does CBD prefer one binding site to the other? Are there orders for CBD binding to these two binding sites? In the cryo-EM data processing, did the author observe different states of Nav1.7, like apo or with only one CBD binding site occupied, were observed? Can the author explain the connection of these two CBD binding site in regulatory of Nav1.7?

Minor points:

1. Fig.3c and Fig.4a, hydrophobic surface will be more informative as CBD is coordinated in these two binding sites by hydrophobic interaction.

2. Line 280: reference for SerialEM is missing.

3. Extended Data Table 1: the model resolution is 2.2 angstrom when FSC equals 0.5?

Revisions made in response to reviews:

We are very grateful to both reviewers for their careful reading of our manuscript, their enthusiastic comments, and especially for their insightful and constructive suggestions for improving the manuscript. We have done new experiments and added new data to the manuscript and we have also revised the presentation and the discussion to address the points that they made. Both the new data and the changes in presentation have improved the manuscript, and we thank the reviewers for their detailed suggestions.

Following is a point-by-point summary of the changes made in the manuscript to address the points raised by the reviewers. In the manuscript, new material is indicated by red text.

Reviewer #1:

The study is a very nice convergence of CryoEM experiments with functional, pharmacological ion channel research. It is fascinating and encouraging to see that methods like cryo-EM are being used as validation tools for experimental results. A physiologically highly relevant question was addressed using cutting edge technical approaches: how does cannabidiol relieve pain? It was shown before that it interacts with Nav1.8, a sodium channel known for its importance in human pain perception. Arguably even more important is Nav1.7, as humans lacking this channel cannot feel pain. The authors show in convincing patch clamp recordings that CBD enhances fast inactivation of Nav1.7. The binding sites are unclear, although the electrophysiological profile hints towards the local anesthetic binding site. To clarify this, the authors present the structure of Nav1.7 bound with CBD and find it in a rather surprising position: close to the IFM motive. Mutagenesis was used to explore this binding site and it was compared to a second binding position which is found closer to the permeation pore on S6 of Domain IV.

The topic is of high interest for both clinicians and basic scientists, as CBD gains more and more therapeutic relevance and the authors describe a new pharmacological binding site for voltage gated sodium channels. The manuscript is well written, the presentation of the data is appropriate. The relevance of each binding site for the channel gating should be investigated in more detail, and it would also allow to exploit the structural data to a greater extent.

Major comments:

- 1. The CryoEM structure shows binding of Beta1 and beta2. The ephys experiments on the other hand were performed in the absence of any beta subunit. Please show that the beta subunits do not interfere with/modify binding of CBD with Nav1.7.*

Good point – we have done new experiments comparing the effect of 1 μ M CBD on both fast inactivation and steady-state inactivation of Nav1.7 with or without co-expression of β 1 and β 2. The shifts of availability induced by CBD with co-expression of β 1 and β 2 are

very similar to those without the beta subunits. We have added a figure showing this data as Supplementary Fig. 10.

2. *Figure 1: The ephys experiments look technically flawless, experiments were performed at 37°C, which is very nice and challenging. Part E: Despite pharmacologically altered recovery kinetics from fast inactivation, the observed slow recovery states of treated channels could also arise from slow inactivated states if the voltage dependence of slow inactivation was altered by CBD. Please discuss and explain, and/or add experimental data.*

We agree, and we have added a sentence stating that the “The slowed recovery from inactivation following 5-s depolarizations to -40 mV (Fig. 1d) could reflect slow recovery of CBD-bound fast-inactivated channels or a larger fraction of channels in the slow inactivated state or a combination of the two.”

3. *The authors show very interesting data on two binding sites of CBD. Especially the I-site is a so far unknown binding site for therapeutics. Therefore, a more detailed functional assessment of the structural-functional relation of each binding site would be desirable. It would be very enriching to further investigate the functional importance for each presented binding sites for the pharmacological effect on Nav1.7. E.g., the authors should use mutagenesis (e.g. to change the fenestration site size) to show the effect of a disruption of this binding site for the functional impact of CBD on the channel, or similar:*

a) Concerning the lipophilic properties of CBD the F Site is likely the more occupied or do the authors assume that the I-side can be accessed via the fenestration pathway? If yes they could also use MD Simulations to investigate and further characterize this pathway.

We have added new data with mutagenesis experiments on the key residues in the F-site, which seemed like the first-order experiments to test the functional importance of this binding site. These mutations resulted in substantial reductions in CBD potency, supporting the functional relevance of this binding site. These new data are shown in Fig. 4c.

Examining the effect of further mutations designed to modify the size of the fenestration is a great idea, and these will be on our list of follow-up experiments.

With regard to the second point, the I-site is located outside of the pore domain, and thus, may be accessible from the lipid bilayer without requiring entry through a fenestration. We have added a sentence clarifying this point. We agree that MD simulations may be interesting to pursue to get insight into movements of CBD between the bilayer to each binding site, but feel these are beyond the scope of the paper and might be best performed in an in-depth manner by groups specializing in such simulations. The structural data in the manuscript should provide a key data set to help guide such simulations.

b) The IV-I fenestration – close to the F-site - was suggested to be one of the smallest fenestrations [Tao and Corry, 2022]. It would be quite interesting and useful if the authors showed some analysis of the side fenestrations with and without the presence of CBD and also compare it to the other known Nav1.7 structures (7W9K, 6J8G).

Thanks for this suggestion. We have examined the IV-I fenestration with and without CBD. In the revised **Supplementary Fig. 4b**, we have illustrated the hydrophobic surface of the IV-I fenestrations in various Nav1.7 structures. Interestingly the structure of the F-site remains nearly unchanged in the presence or absence of CBD. However, the IV-I fenestration is notably diminished in the complex of Nav1.7 bound to saxitoxin and Huwentoxin IV and saxitoxin (PDB: 6J8G). Since Huwentoxin-IV is a gating modifier, and stabilizes the channel in less-activated position, this suggests that the IV-I fenestration may be dynamically regulated during channel gating. We have added a sentence discussing this point.

c) The authors describe certain key residues such as F387 and V383, but do not show any mutagenesis experiments of these residues to determine whether CBD binding is affected (partially, completely or not at all). These may help to show whether the F-site close to the I-IV fenestration may have a higher affinity than or may even be dependent on binding of CBD on the I-Site.

Thanks, this is a very good point. As per the suggestion, we have conducted mutagenesis experiments on the key residues involved in CBD binding at the F-site, F387A and V383A. The results showed that the potency of CBD was substantially diminished by each mutation, supporting the functional relevance of this binding site. These new data are shown in Fig. 4c.

To explore the relative impact of the two binding sites on overall potency of CBD, we have also added data examining the effect of both single and double mutations of S1320A and N1459A in the I-site. These experiments showed a similar reduction in the potency of CBD as the F-site mutations, suggesting that CBD binding at both sites contributes to its overall inhibition. We have shown this new data in Fig. 3e.

d) An earlier study on Nav1.4 showed that the local anesthetic residue F1586 is important for CBD binding to sodium channels and that CBD directly occludes the pore by binding at the local anesthetic site near the DIII-IV fenestration [Ghovanloo et al., 2022]. This contrasts with the conclusion of this paper that CBD probably does not occlude the pore and binds in the IV-I fenestration and IFM binding site. To exclude this, experiments using a mutation of the local anesthetic binding site would help and should be easily performed.

Thanks for this excellent suggestion. As suggested, we investigated the effect of mutating Nav1.7-F1748, which corresponds to the local anesthetic binding site F1586 in Nav1.4. In line with the data in the Ghovanloo paper on Nav1.4, we found that mutating this site in

Nav1.7 did have a modest effect on CBD interaction with the channel, reducing the shift in fast inactivation produced by 1 μ M CBD from 9 to 6 mV. We have added this data as Supplementary Fig. 11 and also added a paragraph to the manuscript discussing this data and referring to the Ghovanloo paper, noting that our results fully support their conclusion that the relatively small effects on CBD inhibition compared to the large effects on local anesthetic inhibition implies that this is not the primary binding site for CBD.

e) The I-site is a very unorthodox site, given how the hydrophobicity of this pocket is considered crucial for proper binding of the motif. Did the authors consider checking for the hydrophobicity of the pocket with and without the drug to see if CBD cause any changes? This analysis should be added to the paper.

We appreciate this insightful point. The data with and without CBD suggest that the the I-site is a new pocket induced by CBD binding. The environment within this pocket is highly hydrophobic. We have included the related hydrophobic surface of the I-site in the revised **Supplementary Fig. 4a**.

f) The authors mention that the F1463 of the IFM motif is in the pocket where CBD binds. Did the authors notice any pi-stack interactions between CBD and this residue? If so, it will help to perform experiments to test if CBD directly interacts with the IFM motif by mutating this residue e.g. to an alanine.

According to the structure, F1473 of the IFM motif appears to be distant from the CBD-binding pocket, whereas I1472 of the IFM motif may be involved in CBD binding at the I-site. This finding is consistent with the results obtained from the binding free energy calculation conducted on F1473A and I1472A mutants.

We have added a figure (**Supplementary Fig. 6**) illustrating the position of the CBD at the I-site relative to F1473 and I1472 and illustrating the calculated changes in binding energies for mutations at these two residues.

Minor Comments:

1. Supplement 4: c and D please change scaling so that quick transient sodium traces become better visible.

Point taken. We have adjusted the scale of the figures in the revised **Supplementary Fig. 8** to improve the visibility of the transient sodium traces.

2. *Depending on the journal's rules, it may be preferable to use SD instead of SEM throughout the manuscript.*

We appreciate the reviewer's suggestion. We have reviewed the guidelines of *Nature Communications* and found no specific requirement for reporting either SD or SEM. Therefore, we have chosen to report the SEM in our data analysis.

3. *Methods: Ephys: No statement on series resistance or voltage error is given (and the pipettes are quite small in diameter). Please mention.*

We have added a sentence giving this information in Methods.

4. *Figure 1: In Figure 1C the relative currents of CBD start from 0.4 but should start from (or close to) 1.*

Thanks - this was confusing because the currents in CBD were plotted relative to the initial current in control in order to illustrate the degree of inhibition by CBD at negative voltages. We have clarified Figure 1C by illustrating the effect of CBD on a representative cell, plotting absolute values of peak current before and after 300 nM CBD, illustrating both the degree of inhibition at negative voltages and the shift of voltage dependence. And we added a dashed line that normalizes the Boltzmann function of the CBD data to the control Boltzmann in order to better illustrate the shift of voltage dependence.

5. *Figure 3(e) and Figure 4(a) inset: Residue interactions are shown only for hydrogen bonds and van der Waal's interactions, but the text in addition to these mentions pi-pi stack interactions which are missing in the figures. Please also show in the figures.*

Thanks for pointing this out. We have modified the related panels in the original **Figure 3** and **Figure 4**, which have been combined as **Supplementary Fig. 5** in the revised manuscript.

Reviewer #2:

The manuscript by Jian and colleagues investigated inhibitory mechanism of human voltage-gated sodium channel Nav1.7 by cannabidiol (CBD) by using structural and electrophysiological approaches. Through electrophysiological studies, the authors confirmed CBD can inhibit activity of Nav1.7 by specifically interacting with Nav1.7 in closed inactivated state rather than closed resting state. The high-resolution single particle cryo-EM complex structure of Nav1.7 with CBD bound at 2.8 Å revealed two distinct binding sites for CBD, one localized in the fenestration in the upper pore, the other close to IFM motif of the intracellular linker between domain III and IV. Structural analysis provided detailed interaction mode of CBD in Nav1.7, and subsequent mutagenesis and functional assay further prove roles of key residues in coordinating CBD. This high-quality work is another important contribution in mechanistic mechanism of the sodium channels field. By unraveling interacting details between Nav1.7 and CBD, this work provides valuable information for future

drug improvement on the basis of CBD, and shed lights on treatment of diseases like epilepsies with little side effects.

Unexpectedly, there are two binding sites observed in the complex structure. These two binding sites differs from binding modes and effects on conformational rearrangement to regulate protein function. *Does CBD prefer one binding site to the other? Are there orders for CBD binding to these two binding sites? In the cryo-EM data processing, did the author observe different states of Nav1.7, like apo or with only one CBD binding site occupied, were observed? Can the author explain the connection of these two CBD binding site in regulatory of Nav1.7?*

The reviewer raises key questions that follow from the results. The static nature of the structural data make it impossible from this data to say whether there is a preference for one binding site over the other. However, we have added new electrophysiological data to the manuscript examining the effects of mutations at the two sites on the potency of CBD inhibition (new Figs 3e and 4c). Interestingly, mutations at the two sites have similar effects to reduce CBD potency, suggesting that both are important for the overall effect of CBD.

We did not observe any apo-state structure during classification even with low-pass filtered apo-state reference. However, as we now illustrate in Supplementary 4, we did observe that the F site with CBD bound remains identical to that in apo-structure. This observation suggests that occupancy of one site is unlikely to be a prerequisite for the other. Of course, a rigorous test of this would be to get structures of CBD binding in channels with each site mutated. This would be a major undertaking and certainly beyond the scope of this initial manuscript.

Minor comments:

1. *Fig.3c and Fig.4a, hydrophobic surface will be more informative as CBD is coordinated in these two binding sites by hydrophobic interaction.*

Thanks - great suggestion. In the revised manuscript, we have utilized the hydrophobic surface presentation to better illustrate the binding environment of both the F-site and I-site in the revised **Figure 3**, **Figure 4**, and **Supplementary Fig. 4**.

2. *Line 280: reference for SerialEM is missing.*

Point taken. The related reference has been added to the revised manuscript.

3. *Supplementary Table 1: the model resolution is 2.2 angstrom when FSC equals 0.5?*

We appreciate the reminder and apologize for the typo. The estimated resolutions by half-map and model-map FCS have been corrected in the revised manuscript.

Reviewers' Comments:

Reviewer #1:

Remarks to the Author:

The authors have addressed the comments well. One minor issue: can you please show the data in Fig. 3f, Suppl Fig.10B and Suppl Fig. 11 lower row as dot plots?

Reviewer #2:

Remarks to the Author:

In the revised manuscript, the authors performed an extensive studies by performing electrophysiological studies of mutants to verify the binding sites observed in structures, and important physiological roles of these key residues play. All these results are consistent to the previous conclusion. In addition, the authors have made sufficient presentation and added more views in discussion to further improve the quality of the manuscript. With all these information in the revised manuscript, most of my concerns raised previously have been addressed in the current version. The authors should consider a minor comment below and modify the manuscript if necessary.

Minor comments:

1. More details should be provided in figure legend of Supplementary Fig. 6.

Reviewer #1:

The authors have addressed the comments well. One minor issue: can you please show the data in Fig. 3f, Suppl Fig. 10B and Suppl Fig. 11 lower row as dot plots?

The ΔV_{50} analysis could not be shown as dot plots. The ΔV_{50} values were firstly obtained by fitting the inactivation curves and presented as mean \pm SEM. Then, the ΔV_{50} values were calculated by " $V_{50,experimental\ group} - V_{50,control\ group}$ " and the uncertainties of ΔV_{50} s were calculated under the law of propagation of uncertainties: " $\sigma =$

$$\sqrt{\sigma_{experimental\ group}^2 + \sigma_{control\ group}^2} \cdot$$

Reviewer #2:

In the revised manuscript, the authors performed an extensive studies by performing electrophysiological studies of mutants to verify the binding sites observed in structures, and important physiological roles of these key residues play. All these results are consistent to the previous conclusion. In addition, the authors have made sufficient presentation and added more views in discussion to further improve the quality of the manuscript. With all these information in the revised manuscript, most of my concerns raised previously have been addressed in the current version. The authors should consider a minor comment below and modify the manuscript if necessary.

Minor comments:

1. *More details should be provided in figure legend of Supplementary Fig. 6.*

Point taken. We have modified the legend for **Supplementary Fig. 6** to provide a more concise and descriptive explanation in the revised manuscript "**Supplementary Fig. 6 | Position of CBD in the I-site relative to the position of the IFM wedge. In silico alanine scanning indicates that Ile1472 might play a role in CBD binding at the I-site. Positive values of relative binding energies, calculated by the Prime-MM/GBSA method, indicate that substituting alanine for the native residue results in less favorable interactions with CBD.**"